# Performance and Durability of Non-Stick Coatings Applied to Stainless Steel: Subtractive vs. Additive Manufacturing

**DOI:** 10.3390/ma16175851

**Published:** 2023-08-26

**Authors:** Guillermo Guerrero-Vacas, Oscar Rodríguez-Alabanda, Francisco de Sales Martín-Fernández, María Jesús Martín-Sánchez

**Affiliations:** 1Department of Mechanical Engineering, Higher Polytechnic School, University of Córdoba, Rabanales University Campus, 14014 Córdoba, Spain; orodriguez@uco.es; 2Department of Civil, Materials and Manufacturing Engineering, School of Industrial Engineering, University of Malaga, 29071 Malaga, Spain; fdmartin@uma.es

**Keywords:** anti-adherent coatings, stainless steel, conventional manufacturing, additive manufacturing, adhesion, substrates

## Abstract

This study compares subtractive manufacturing (SM) and additive manufacturing (AM) techniques in the production of stainless-steel parts with non-stick coatings. While subtractive manufacturing involves the machining of rolled products, additive manufacturing employs the FFF (fused filament fabrication) technique with metal filament and sintering. The applied non-stick coatings are commercially available and are manually sprayed with a spray gun, followed by a curing process. They are an FEP (fluorinated ethylene propylene)-based coating and a sol–gel ceramic coating. Key properties such as surface roughness, water droplet sliding angle, adhesion to the substrate and wear resistance were examined using abrasive blasting techniques. In the additive manufacturing process, a higher roughness of the samples was detected. In terms of sliding angle, variations were observed in the FEP-based coatings and no variations were observed in the ceramic coatings, with a slight increase for FEP in AM. In terms of adhesion to the substrate, the ceramic coatings applied in the additive process showed a superior behavior to that of subtractive manufacturing. On the other hand, FEP coatings showed comparable results for both techniques. In the wear resistance test, ceramic coatings outperformed FEP coatings for both techniques. In summary, additive manufacturing of non-stick coatings on stainless steel showed remarkable advantages in terms of roughness, adhesion and wear resistance compared to the conventional manufacturing approach. These results are of relevance in fields such as medicine, food industry, chemical industry and marine applications.

## 1. Introduction

Non-stick coatings are highly important in various industrial sectors, such as the food industry [1], aerospace, automotives, the metal transformation industry [2,3] and the chemical industry. These coatings improve the efficiency of processes, reduce friction and demolding, prevent the adhesion of all kinds of masses and liquids and prevent the accumulation of residues, among other advantages. It is common to apply non-stick coatings to stainless-steel elements, such as equipment elements that handle food masses, as well as elements and parts of chemical reactors and molds for forming polymeric parts. Other types of coatings have been used on stainless steel for different purposes, such as fluorooxysilane for anti-icing applications [4], NbC coatings for proton exchange membrane fuel cells (PEMFC) [5], MnCo_2_O_4_ coatings on solid oxide fuel cells (SOFC) [6], TiO_2_ coatings for biomedical applications [7] and ceramic and graphene coatings for corrosion protection of stainless steel [8]. Coatings are usually applied on substrates obtained by conventional manufacturing processes such as machining, but now with the implementation of new manufacturing technologies such as metal 3D printing, it is necessary to understand the differences they present. Now, there is a growing interest in additive manufacturing (AM). This, also known as 3D printing, has gained popularity in various fields of industry due to its ability to produce complex components quickly and efficiently. In the literature, we found works that investigated the adhesion of other types of coatings on conventional stainless-steel substrates, for example, with the aim of minimizing biological fouling on ship hulls and sensors [9,10] or to improve the antimicrobial effect of surfaces in contact with food [11]; however, no specific studies of non-stick coatings on substrates obtained by AM are known.

In recent years, significant attention has been paid to the study of the evaluation and bonding procedures of polymeric and ceramic coatings on metal substrates. Among the investigations that have been carried out, we investigated the behaviors of steel–epoxy resin [12], titanium alloy–PEDOT coating [13], aluminum and polypropylene [14] and stainless steel–MoB/CoCr coating [15]. All these works evaluated the influence of surface roughness, the interactions between polymers and metals and the methods to generate these textures. On the other hand, there are some works that study the influences of the formation of coatings in pieces obtained by AM. This is the case of HVOF (High-Velocity Oxygen-Fuel) cold thermal spraying of ceramic–metallic particles (cermet) on 3D printed elements of Cu, Ti and Mg [16] and also of polymeric parts obtained by 3D printing coated with hydroxyapatite [17]; further, an exhaustive work on the limitations, applications and advantages of using various additive manufacturing techniques to make metallic coatings was found [18]. However, there are no known studies that address the evaluation of non-stick coatings applied by spray painting with a manual spray gun to metal parts obtained by additive manufacturing.

In this work, two types of coatings representative of the two types of coatings present in the specialized industry are selected for evaluation, namely coatings of fluoropolymeric origin, such as coatings with FEP [19] and a siloxane-based non-stick ceramic coating obtained by sol–gel [20].

The objective of this study is to compare the anchoring differences, wear resistance and wettability of two types of non-stick coating on stainless-steel substrates obtained by conventional manufacturing and by additive manufacturing. We intend to analyze and evaluate the adhesion of the non-stick coatings to substrates. In addition, the impact of pre-treatments on substrates, surface roughness and wettability is investigated. By comparing the results obtained in the abrasive particle projection tests and in the lattice cutting test, the aim is to identify the best adhesion conditions of the coatings to the substrates in both production methods.

## 2. Materials and Methods

### 2.1. Specimens

A set of 52 unit specimens were produced, each with dimensions of 30 × 30 × 5 mm. Among these 52 specimens, 26 units were manufactured using cutting and machining processes, referred to as SM (subtractive manufacturing). The remaining 26 specimens were produced using additive manufacturing (AM). Various scenarios were explored, as outlined in Table 1. These scenarios included the specimens in their initial state from the machining process, the specimens from the additive manufacturing equipment, the specimens after undergoing particle blasting treatment (shot blasting) and, finally, the specimens with the studied coatings.

Figure 1 includes images of the specimens obtained by both techniques, before and after being treated.

In this study, the same type of stainless steel was used to prepare the specimens: stainless steel DIN EN 10088:2015–1.4542 [21], also known as AISI 630, both for the specimens obtained by AM and for specimens processed by flat milling (SM).

The steel used for both processes had a similar composition and was hardenable and martensitic with outstanding properties such as high wear resistance, good corrosion resistance and high elastic limit. Applications include screws, construction, the chemical and pharmaceutical industry, spindles and nuclear engineering, among others.

In the case of AM, the material was supplied in the form of filament spools agglomerated from metal powder, polymeric binder and waxes. For the other type of substrate, the specimens were laminated, cut and, finally, machined by milling. Table 2 shows the nominal composition of the steel used.

The specimens obtained by machining were extracted from a 150 mm × 200 mm × 25 mm laminated plate. Subsequently, they were cut with a FAT 330 band saw (FAT Soluciones de Corte S.L.U, Barcelona, Spain) and machined on a Chevalier QP 2026-L machining center (Falcon Machine Tools Co., Ltd., Shengang, Taiwan). A Ø 50 mm milling head with 8 uncoated carbide inserts was used at a cutting speed of 2000 rpm, with a feed rate *f* = 100 mm/min, a depth of pass a_p_ = 0.5 mm and a_e_ = 40 mm. The sharp edges were then filed down and marked.

The specimens obtained by AM were made using the Metal X equipment from Markforged (Watertown, MA, USA), by means of fused filament deposition (FFF) with atomic diffusion technology (ADAM), consisting of a 3D printer, washing station (Wash-1) and sintering furnace (Sinter-1). The material (DIN EN 10088:2015–1.4542 stainless steel) was agglomerated in a polymer matrix, in the form of a 1.75 mm diameter filament, using an extrusion nozzle of 0.4 mm diameter and a layer height of 0.125 mm. Each of the specimens was configured with a triangular deposition pattern fill and four outer solid layers, with zig-zag deposition (upper, lower and wall thickness layers).

The accompanying figure (Figure 2) shows images from confocal microscopy of both the SM specimen (left) and the AM specimen (right).

The additively manufactured specimens, after passing through a washing station where the binder of the filament was substantially eliminated, were sintered in an oven with an inert atmosphere of argon and a mixture of argon and hydrogen. It must be noted that the specimens were oversized in the 3D printing process (35.9 mm × 35.9 mm × 6.0 mm) so that, after washing and sintering, the contraction that occurred was compensated, obtaining the required final dimensions (30.0 mm × 30.0 mm × 5.0 mm).

Subsequently, a subset of specimens obtained by these techniques was treated by abrasive spraying with brown corundum (MPA, Cornellá de Llobregat, Barcelona, Spain) at 0.4 MPa for the pretreatment of the substrates and the application of the coatings.

### 2.2. Coatings

The non-stick coatings used in this study were the following: (i) Tecnimacor TF 8840, an FEP-based fluoropolymer supplied by Whitford S.R.L (Brescia, Italy), and (ii) Tecnimacor Califal of sol–gel type, supplied by PPG Industrial Coatings (Pittsburgh, PE, USA). Both coatings were applied by the specialized company Tecnimacor (Tecnimacor S.L, Córdoba, Spain). The coatings were applied by a spray-paint system with HVLP (high-volume low-pressure)-type equipment and subsequently cured, in the case of FEP at 380 °C and for ceramics at 300 °C. To determine the thickness of the coatings, Karl Deutsch equipment, model Leptoskop 2042 (Karl Deutsch, Wuppertal, Germany), was used, which is a portable piece of equipment that uses the induced current method with an uncertainty of ±1 μm. Five measurements were made at different points of the analyzed surface.

A Mitutoyo SJ-201 roughness tester (Mitutoyo Corporation, Kakatsu-ku, Kawasaki, Kanagawa, Japan) allowed us to obtain the roughness parameters Ra (average roughness) and Rz (maximum roughness) on the surfaces of the specimens. So, 4 measurements were made per specimen, in positions perpendicular to each of the edges. The standard used was ISO 21920-2:2021 [22].

Regarding wettability, the sliding angle (SA) were measured. The sliding angle assesses the ability of a liquid to slide on a solid surface. Sliding angle is a measure of the ease with which a liquid droplet can move down a sloping surface before it begins to slide or break away [23,24]. It is an important property in applications such as water-repellent coatings, self-cleaning surfaces and, by similarity, non-stick coatings [25,26]. The sliding angle shown by the surface in contact with the water was measured with a turntable equipped with a digital goniometer of 0.05° appreciation. The platform rotated at a rate of 1°/s. The volume of the deposited drops was 100 μL of purified and deionized water. The measurement was repeated 4 times at various representative points of the studied surface.

Finally, 3D images were captured and processed with Leica DVM6 and DCM8 digital confocal microscopes (Leica microsystems, Wetzlar, Germany). Table 3 includes some characteristics of the coatings used in this study.

### 2.3. Tests for Evaluation of Adhesion to the Substrate: Cross-Cut Adhesion Test (ISO 2409-2020)

A substrate adhesion test was carried out with a TQC/SP300 (TQC Sheen BV, Amsterdam, The Netherlands) using the ISO 2409-2020 standard [27]. Basically, the standard stipulates making cuts in the coating down to the substrate, forming a square lattice, and then fixing an adhesive tape over the cut area to assess the effect of detachment of the coating from the substrate. Using a suitable cutting blade and a metal plate with openings for the spacing of the incisions in the form of a square grid, the cuts established in the standard were made. After the adhesive tape was fixed to the cut area and removed, a coating classification from 0 to 5 was obtained according to the specifications of the standard in such a way that the value 0 indicated that no square was detached from the grids formed and 5 indicated that all the squares were detached. In addition, the percentage of affected area, i.e., the area detached on the adhesive tape, was evaluated. Three different areas of the coating were evaluated. This test was used to evaluate the adhesion strength between a layer of paint or coating and the substrate to which it is applied [28].

### 2.4. Evaluation of Coating Wear

The abrasive projection test, also known as abrasive blasting test or shot-blasting test, is a method used to evaluate the resistance of a material or coating to abrasion [29] but also to analyze the response of the coating to anchoring to the substrate [30]. The behavior of different substrates in the abrasive blasting test can vary depending on their hardness, chemical composition, microscopic structure, surface treatments and the type of abrasive used in the test. Two abrasives were used in this test: glass microspheres and walnut shells. On the one hand, glass microspheres were selected because this abrasive produces impact wear on the surface of the coating, leaving a spherical footprint equivalent to a soft burnishing [31] and, on the other hand, walnut shells were selected because the wear is of the shearing type and generates micro-cuts [32]. In short, we wanted to study the response of the coatings and their relationship with the substrates through interactions with different abrasion phenomena: impact and shearing. In addition, the choice was intentional since they are known to produce wear that allows progressive comparison of the effects. In this way, the specimens were subjected to blasting with Sandblast Cabinet CAT900 equipment from Metalworks (Aslak S.L, San Quirce del Vallés, Barcelona, Spain). Walnut shell abrasive and glass microspheres were supplied by MPA (Cornellá de Llobregat, Barcelona, Spain). A pressure of 0.4 MPa was used with a Ø 6.5 mm nozzle at a distance between the end of the blast gun and the substrate of 200 mm. The characteristics of the abrasives used are shown in Table 4.

## 3. Results and Discussion

### 3.1. Surface Roughness

The roughness of the specimens was studied in different states: specimens directly after machining, SM or AM processes; specimens after surface preparation by spraying particles with brown corundum at 0.4 MPa; and specimens once the FEP and ceramic-type base coatings were applied. In Figure 3, the states studied are schematized. The values obtained from the surface roughness are shown in Figure 4.

It is observed that the surface roughness of the specimens obtained by additive manufacturing (AM) is higher than those obtained by subtractive manufacturing (SM) in phase I before blasting and the Ra values are more than four times higher in the specimens of AM compared to those of SM (2.65 vs. 0.59 μm). After blasting in phase II, the roughness values increase and tend to equalize between the two cases, with Ra values between 2.9–3.5 μm.

However, slightly higher values are still observed in the AM specimens compared to those of SM. After the application of non-stick coatings, both FEP-based and ceramic, a significant reduction in roughness is achieved; for AM, Ra is close to 1 µm for both coatings and phases and in SM, for FEP, it ranges between 0.5–0.9 µm. This is due to the smoothing effect that occurs when applying a homogeneous layer of coating. Although the difference in roughness between the AM and SM specimens after the application of coatings, phase II+ coating, is slightly higher in the AM specimens, it is not very representative. Finally, it is observed that the ceramic coating could not be applied correctly to the SM specimens in phase I, since it did not anchor properly. This was to be expected since among the applicability conditions for this coating, it is indicated that a minimum Ra value of 2 µm is necessary.

### 3.2. Sliding Angle

The sliding angle was evaluated using 100 µL water drops. The drops are of a relatively large size but were selected with the objective of evaluating surfaces of an industrial nature that interact with real drops in that size environment. The accompanying figure (Figure 5) shows the results obtained.

Specimens of phases I and II, that is, uncoated substrates, are not represented in Figure 4; in both cases, the values of SA (sliding angle) exceed 40° and they are hydrophilic surfaces. In phase I+ coating, it is observed that the substrates obtained by SM present lower values than those obtained by AM: 9.8 vs. 14.5° for the case of FEP and values of 17.8° for the ceramic coating by AM since, in SM, the coating does not achieve fixation. In phase II and after the blasting procedures, the SA values decrease; in the case of FEP, the jump is 3° for AM alone; in the case of ceramic coating, the jump is of the same order, 3.2°. In the case of machined specimens, the jump is of a lesser magnitude, 1.1°, in the coatings with FEP. The ceramic coating on the machined specimen, as has already been said, failed, that is, it did not have any anchorage and we found ourselves with bare metal without a coating.

On the other hand, it can be observed that the SA values are higher in AM than SM for the FEP coating. For the ceramic coating, the values after blasting are almost indistinguishable regardless of whether the specimen is obtained by SM or AM.

We postulate that the observed differences in sliding angles between the coatings applied on the two types of substrate appear to derive from subtle differences in surface topography, in the micrometer range. Figure 4 and Figure 5 illustrate these differences visually, where more pronounced roughness correlates with higher sliding angles.

### 3.3. Cross-Cut Adhesion Test

The mesh cut test, also known as the cut adhesion test, yields the results in Table 5.

Regarding the ceramic coverings in phase I, a significant difference is observed between AM and SM. In AM, the samples present a value according to ISO 2409 of 3, indicating a reasonable degree of adherence of the coating. The percentage of affected area during the cross-cut adhesion test was 23%. In contrast, in SM, the samples obtained provide a value of 6 according to ISO 2409, which indicates a total detachment of the coating since the percentage of affected area is 100%, which suggests a total lack of adherence of the ceramic coating to the specimens made by subtractive manufacturing.

For the FEP coatings, differences were again observed between AM and SM. In AM, both the samples with phase I or phase II coating obtained values of 3 and 0, respectively, according to ISO 2409. The percentage of affected area in the AM samples was 30% for coatings in the supplied state and 0% for samples exposed to blasting (phase II). In the case of SM, the samples in the supplied state obtained a value of 4, according to ISO 2409, indicating poor adhesion. The percentage of affected area was 71%. However, in the SM samples subjected to blasting, no affected areas appeared, indicating a perfect anchorage of the coating.

In the two states analyzed, we can say that in phase I (without blasting), ceramic coatings have a level of anchorage to the substrate obtained by AM of the same order as FEP and an affected area of 20–30% in both cases. For substrates obtained by SM, ceramic coatings fail (100% affected area) and FEP coatings have an affected area of 71%, somewhat less. In phase II, ceramic coatings have a perfect anchorage level for both AM and SM in the same way as FEP coatings.

In short, we can indicate that the manufacture of the specimens by additive manufacturing (AM) for the coatings studied performs as well as, or in some cases better than, the specimens obtained by subtractive manufacturing (SM) from the point of view of anchorage to the substrate. On the other hand, it is also observed that FEP-type coatings are more robust and less sensitive than ceramic coatings from the point of view of substrate manufacturing procedures.

### 3.4. Wear Test by Abrasive Particle Projection

In the case of the FEP coating, photographs were taken of the samples obtained by AM and SM both in phase I and phase II and worn with walnut shells. The results are shown in Figure 6. Figure 7 shows the results obtained for the ceramic coating.

Considering the images obtained, walnut shell wear produces less coating detachment in the substrates obtained by AM compared to those of SM for the FEP-based coating. It is observed that there are still some traces of coating in the damaged area in the case of AM, but nothing in the case of SM. In any case, to guarantee a higher-quality anchorage, prior sandblasting of the substrate is necessary in both cases (phase II).

In the case of ceramic coating, the results are in the same order. Substrate coatings by AM are slightly better than SM. On the other hand, it can be observed that ceramic coatings are more resistant to wear than those of the FEP type, since once they are applied on substrates prepared by blasting (phase II), practically no modifications are observed.

The results obtained with the abrasive glass microspheres are shown in Figure 8 and Figure 9.

As indicated, the abrasive with glass microspheres was extremely efficient from the point of view of removing ceramic coatings and within a very short exposure time, it was almost eliminated. In short, there are no significant differences in the anchorage of the coating between the substrates obtained by AM or SM techniques because of wear due to impact in ceramic coatings.

In short, for wear effects due to impact phenomena in non-stick coatings on substrates obtained by additive manufacturing (AM) or subtractive manufacturing (SM), there are no differences.

Among the coatings analyzed, and without either of the two studied being particularly efficient at withstanding wear due to impact effects, which is produced mainly by the abrasive glass microspheres, the FEP coating seems to perform better than the ceramic coating, which, being harder, is more fragile.

## 4. Conclusions

The following conclusions were made:-Roughness (Ra): The specimens obtained by additive manufacturing (AM) present a higher surface roughness compared to those obtained by subtractive manufacturing (SM) in phase I, i.e., without shot blasting, i.e., as obtained from forming processes (Ra of 2 vs. 0.5 μm). After shot blasting in phase II, the roughness values tend to equalize and increase. However, slightly higher values are still observed for AM vs. SM specimens (increase of 0.2–0.3 μm). After the application of anti-adhesion coatings, a significant reduction in roughness is achieved in both cases, obtaining final Ra values between 0.6 to 1 μm, although a slight difference persists, with higher values, in the AM specimens.-Sliding angle (SA): The SA values are higher, indicating worse wettability, in the AM specimens than in the SM specimens for both phase I and phase II FEP coatings (11.5° vs. 8.7° in phase II). However, in the case of the ceramic coating, the values are practically identical after surface blasting, regardless of whether the specimens were obtained by AM or SM (≈15°).-Cross-cut adhesion test: Significant differences are observed in the adhesion of the specimens with ceramic coating between AM (23% affected area) and SM (100% affected area) in phase I. In phase II, the results are equal. In the case of FEP coatings, adhesion results are better for AM vs. SM in phase I (30% affected area vs. 70%) and in phase II, they are similar for AM and SM.-Abrasion wear: It is observed that substrates obtained by AM show slightly higher values (less wear) in shear wear compared to SM, especially for FEP-based coatings, but also in the case of ceramic coating, both in phase I and phase II. On the other hand, wear by impact phenomena in anti-adhesion coatings (FEP and ceramic) on substrates obtained by additive manufacturing (AM) or subtractive manufacturing (SM) shows no differences.

This study compares non-stick coatings on stainless steel by conventional and additive manufacturing. In general, additive manufacturing (AM) performs better in shear wear and anchorage to the substrate of FEP fluoropolymer and sol–gel ceramic coatings than shear and machining procedures, i.e., subtractive manufacturing (SM). AM has slightly higher roughness and wettability (worse wettability) than SM for FEP coatings, but not for ceramic coatings.

Finally, it can be suggested that additive manufacturing (AM) of stainless-steel substrates is a suitable option with better properties for sol–gel ceramic coatings when they are applied as part of a complete process, i.e., additive manufacturing and blasting, compared to subtractive manufacturing. On the other hand, in the case of FEP-type fluoropolymer coatings applied on stainless-steel substrates, the results indicate that additive manufacturing (AM) is favorable regarding anchorage to the substrate and wear, suffering a slight loss in wettability properties, compared to subtractive manufacturing (SM).

## Figures and Tables

**Figure 1 materials-16-05851-f001:**
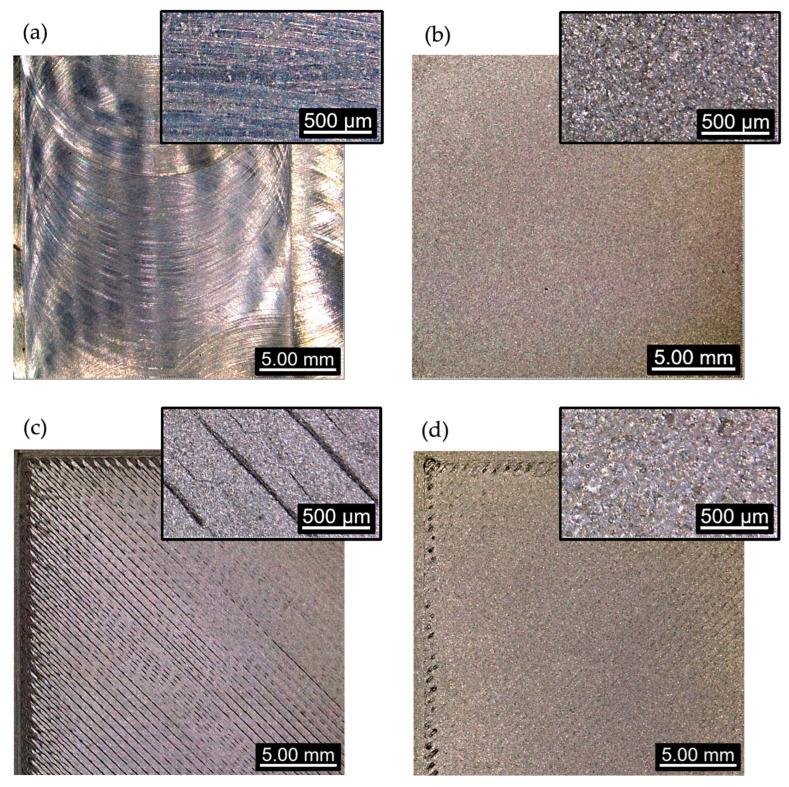
Images obtained with a Leica DVM6 confocal microscope on specimens: (**a**) subtractive manufacturing (SM), (**b**) SM + blasting, (**c**) additive manufacturing (AM), (**d**) AM + blasting.

**Figure 2 materials-16-05851-f002:**
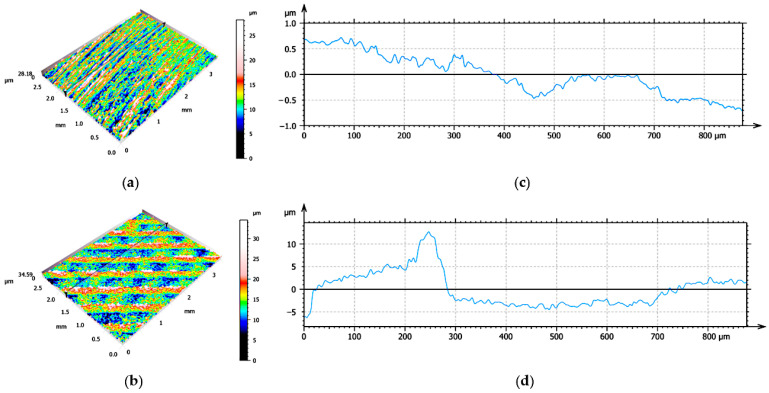
Images obtained with a Leica DCM8 confocal microscope and treated with the Mountain v8 software on specimens obtained by (**a**) subtractive manufacturing (SM), 5× objective, (**b**) additive manufacturing (AM), 5× objective, (**c**) surface profile after section transverse to the specimen obtained by SM, 20× objective, (**d**) profile of the surface after transversal section to the specimen obtained by AM, 20× objective.

**Figure 3 materials-16-05851-f003:**
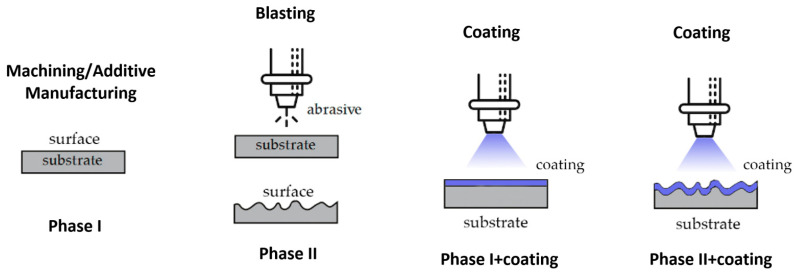
The states in which the surface roughness was studied.

**Figure 4 materials-16-05851-f004:**
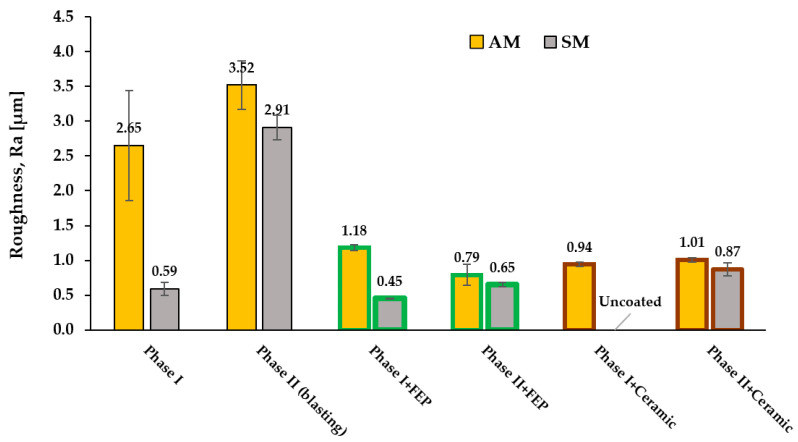
Mean roughness Ra after manufacturing, surface treatment and application of FEP fluoropolymeric coating/ceramic coating (green and red colors represent FEP an ceramic coatings).

**Figure 5 materials-16-05851-f005:**
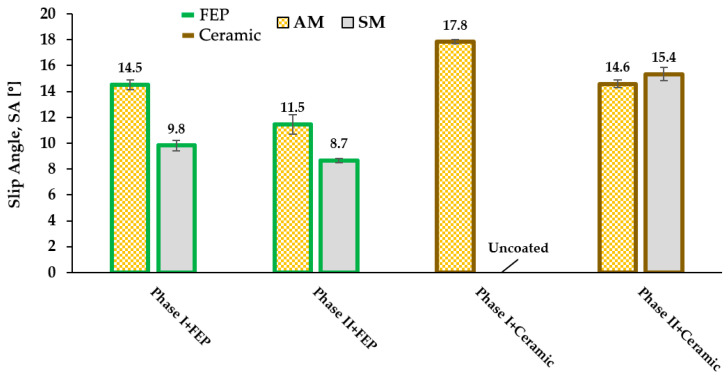
Sliding angle SA with 100 µL drops after manufacturing, surface treatment and application of FEP fluoropolymeric coating/ceramic coating.

**Figure 6 materials-16-05851-f006:**
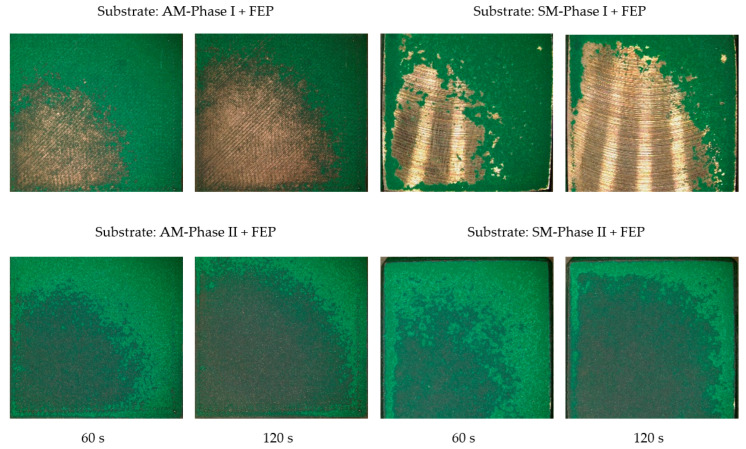
Images obtained with a Leica DVM6 12× microscope of specimens obtained by AM and SM after projection of the walnut shell abrasive on the FEP coating at 0.4 MPa.

**Figure 7 materials-16-05851-f007:**
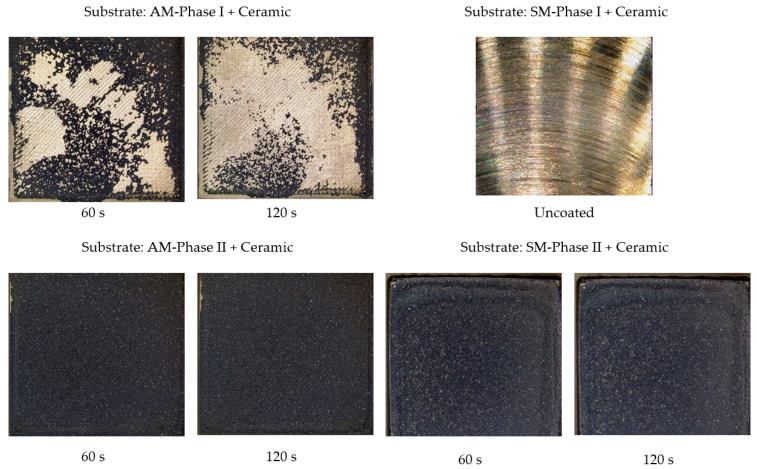
Images obtained with a Leica DVM6 12× microscope of specimens obtained by AM and SM after projection of the walnut shell abrasive on the ceramic coating at 0.4 MPa.

**Figure 8 materials-16-05851-f008:**
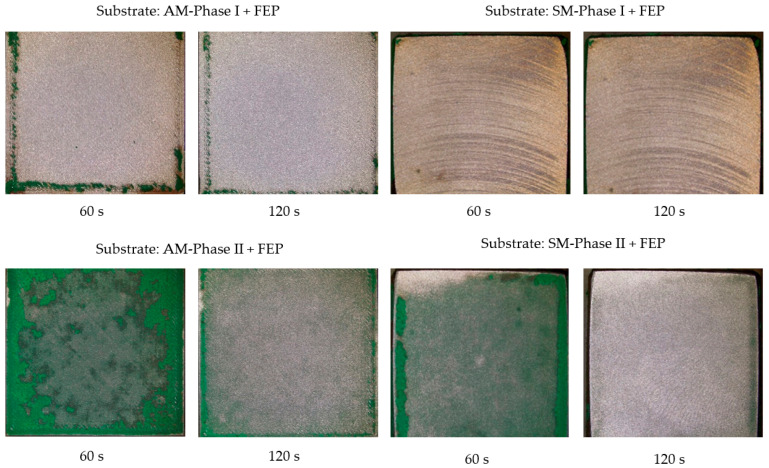
Images obtained with a Leica DVM6 12× microscope of specimens obtained by AM and SM after projection of the abrasive glass microspheres on the FEP coating at 0.4 MPa.

**Figure 9 materials-16-05851-f009:**
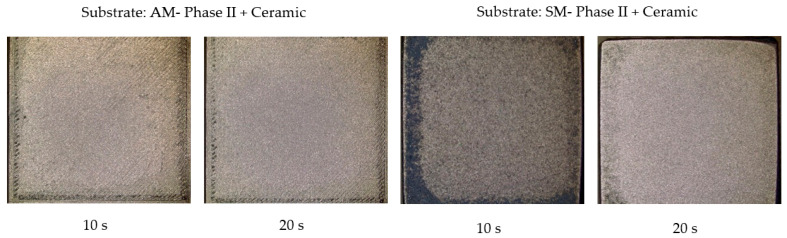
Images obtained with a Leica DVM6 12× microscope of specimens obtained by AM and SM after projection of the abrasive glass microspheres on the ceramic coating at 0.4 MPa.

**Table 1 materials-16-05851-t001:** Phases and number of specimens used in the experiments.

	AM	AM + Blasting	SM	SM + Blasting
Uncoated	3	3	3	3
FEP Coating	5	5	5	5
Ceramic Coating	5	5	5	5

**Table 2 materials-16-05851-t002:** Chemical composition of DIN EN 10,088–1.4542 stainless steel.

Substrate	Cr	Ni	Cu	Nb + Ta	Mn	Si	C	P	Co	S	Fe
DIN EN 10088–1.4542	15–17%	3–5%	3–5%	0.15–0.45%	<1%	<1%	<0.07%	<0.04%	<0.10%	<0.03%	Rest

**Table 3 materials-16-05851-t003:** Main characteristics of Tecnimacor 8840 and Tecnimacor Califal coatings.

Reference	Resin	RAL Color	Coating Thickness (μm)
8840	FEP	140 40 30	32.60 ± 2.54
Califal	Siloxan sol–gel	000 20 00	27.53 ± 1.27

**Table 4 materials-16-05851-t004:** Main characteristics of abrasives used for wear experiments.

Abrasive	Mohs Hardness	Grain Size (µm)	Specific Weight (g/cm^3^)
Glass microspheres	6	200–300	2.5
Walnut shells	2.5–3	450–1000	1.2–1.4

**Table 5 materials-16-05851-t005:** Classification according to ISO 2409 and % of affected area.

Manufacturing Procedure	Phase + Coating	ISO 2409	% Affected Area
AM	I + Ceramic	3	23
AM	II + Ceramic	0	0
SM	I + Ceramic	5	100
SM	II + Ceramic	0	0
AM	I + FEP	3	30
AM	II + FEP	0	0
SM	I + FEP	4	71
SM	II + FEP	0	0

## Data Availability

Data can be provided by Guillermo Guerrero-Vacas at guillermo.guerrero@uco.es.

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
