# Peer review of "Performance and Durability of Non-Stick Coatings Applied to Stainless Steel: Subtractive vs. Additive Manufacturing"

_materials, 2023, doi:10.3390/ma16175851_

Round 1

Reviewer 1 Report

Dear author

Please do the notes contained in the attached report.......best regard

Author Response

Los autores desean agradecer al revisor por las sugerencias de mejora y los comentarios reflexivos para mejorar el trabajo.

Reviewer 2 Report

1-What kind of additive manufacturing is used in this work? This is missing in the abstract.

2-Even what type of conventional manufacturing is used for coating in this work?

3-Authors must write in the abstract; what kind of wear testing have they used?

4- For the paragraph in the introduction starting with line 57; the authors must introduce some other coatings that are used on Stainless Steel with specific purpose.

 5-Even authors jumped very early on the additive manufacturing in the introduction without discussing much about the conventional coating processes on SS.

6- The first paragraph is confusing, the 26 units were cut and machined by subtractive and 26 were prepared by additive. The authors need to revise the title like additive or subtractive manufactured specimens of SS.

7- kindly include the macro images of specimens prepared through SM and AM under heading 2.1. for better understanding.

8- Usually the surface roughness of AM-prepared specimens is high, so this goes positively in the direction of coating or not?

9- Why slip angle is important to analyze?

10- The heading 3.3. and 3.4. comes under the single umbrella of wear testing. Even though the description of these headings is not enough, the authors need to extend it as this is an important result to counter the quality assessment of coating.

11- Authors need to rewrite the conclusion must on the standardized practice. The last three paragraphs starting from line 354 are quite vague and may be shifted to some other section. Also, the conclusion must be written on some quantitative values.

minor changes 

Author Response

The authors would like to thank the reviewer for the suggestions for improvement and thoughtful comments to improve the work.

Reviewer 3 Report

The manuscript titled " Performance and Durability of Nonstick Coatings Applied to Stainless Steel: Subtractive vs. Additive Manufacturing" would be an interesting fit for publication in Materials but the article would need major revision for the following comments before publication.

  1. In the introduction section of the manuscript, the authors should elaborate more on the disadvantages of crack formation in FEP and ceramic-based fluoropolymeric coatings deposited by sol-gel process because of the limitation of the maximum film thickness achievable without crack formation via single step deposition.
  1. In the methodology section, it is recommended to perform Atomic Force Microscopy of the coatings. Such a study would be insightful in determining the surface rough and elastic properties of the coatings deposited by AM vs SM.
  1. The authors should give some evidence of formation of the FEP and ceramic-based fluoropolymeric coatings on stainless steel. In order to understand the composition and purity of the coatings, a XRD study is needed.
  1. Please discuss in detail about the impact of surface tension and adsorption properties of the coatings produced by AM vs SM on the slip angle.
  2. The coatings would need further investigation under SEM to understand their morphology, pitting, and grain boundary composition.

Dear Editor,

The manuscript titled " Performance and Durability of Nonstick Coatings Applied to Stainless Steel: Subtractive vs. Additive Manufacturing" would be an interesting fit for publication in Materials but the article would need major revision for the following comments before publication.

Thanks!

Author Response

(The authors gave the same response as above.)

Round 2

Reviewer 3 Report

The authors have addressed all the comments/concerns in the revised version of the manuscript and the rebuttal letter, it is recommended for publication.